# A Pattern-Recognition-Based Ensemble Data Imputation Framework for Sensors from Building Energy Systems

**DOI:** 10.3390/s20205947

**Published:** 2020-10-21

**Authors:** Liang Zhang

**Affiliations:** National Renewable Energy Laboratory, Buildings and Thermal Sciences Center, Golden, CO 80401, USA; Liang.Zhang@nrel.gov

**Keywords:** missing data, data imputation, ensemble method, pattern recognition, machine learning, building sensors

## Abstract

Building operation data are important for monitoring, analysis, modeling, and control of building energy systems. However, missing data is one of the major data quality issues, making data imputation techniques become increasingly important. There are two key research gaps for missing sensor data imputation in buildings: the lack of customized and automated imputation methodology, and the difficulty of the validation of data imputation methods. In this paper, a framework is developed to address these two gaps. First, a validation data generation module is developed based on pattern recognition to create a validation dataset to quantify the performance of data imputation methods. Second, a pool of data imputation methods is tested under the validation dataset to find an optimal single imputation method for each sensor, which is termed as an ensemble method. The method can reflect the specific mechanism and randomness of missing data from each sensor. The effectiveness of the framework is demonstrated by 18 sensors from a real campus building. The overall accuracy of data imputation for those sensors improves by 18.2% on average compared with the best single data imputation method.

## 1. Introduction

It is reported by U.S. Energy Information Administration that buildings consumed 32% of primary energy in 2019 in the United States and global energy consumption in buildings will grow by 1.3% per year on average from 2018 to 2050 [1]. Building operation data are playing an important role in building design, retrofit, commissioning, maintenance, operations, monitoring, analysis, modeling, and control. With the wide adoption of building automation system (BAS), smart sensors, and Internet of Things in buildings, massive measurements from sensors and meters are continuously collected, providing a +great amount of data on equipment and building operations and great opportunities for data-driven tools to improve building energy efficiency based on collected data [2].

Data quality is essential for data-driven tools and missing data is one of the most common and important issues for data quality. During building operations, it is common for sensors to fail to record data for several reasons, including malfunctioning equipment or sensors, power outage at the sensor’s node, random occurrences of local interferences, and a higher bit error rate of the wireless radio transmissions as compared with wired communications [3]. Cabrera and Zareipour [4] summarized that missing data can be grouped depending on the reason they are missing: (a) missing not at random, or systematic missing: when the probability of an observation being missing depends on the information that is not observed; (b) missing at random: when the probability of an observation being missing depends on other observed values; and (c) missing completely at random: when the probability of an observation being missing is completely at random and not related to any other value.

In statistics, imputation is the process of replacing missing data with substituted values [5]. The theories of imputation are well studied. Early in 2001, Pigott [6] wrote a review of methods for missing data imputation and he summarized that ad hoc methods, such as complete case analysis, available case analysis (pairwise deletion), or single-value imputation, are widely studied. Ad hoc methods can be easily implemented, but they require assumptions about the data that rarely hold in practice. However, model-based methods, such as maximum likelihood using the expectation-maximization algorithm and multiple imputation, hold more promise for dealing with difficulties caused by missing data. While model-based methods require specialized computer programs and assumptions about the nature of the missing data, these methods are appropriate for a wider range of situations than the more commonly used ad hoc methods. Harel and Zhou [7] reviewed some key theoretical ideas, forming the basis of multiple imputation and its implementation, provided a limited software availability list detailing the main purpose of each package, and illustrated by example the practical implementations of multiple imputation, dealing with categorical missing data. Ibrahim et al. [8] reviewed four common approaches for inference in generalized linear models with missing covariate data: maximum likelihood, multiple imputation, fully Bayesian, and weighted estimating equations. They studied how these four methodologies are related, the properties of each approach, the advantages and disadvantages of each methodology, and computational implementation. They examined data that are missing at random and nonignorable missing. They used a real dataset and a detailed simulation study to compare the four methods.

Many existing studies are found to apply data imputation in missing data in buildings. Deleting the missing data is the most direct way to deal with them and, on some occasions, it can be counted as a data imputation method. In the research from Ekwevugbe et al. [9], all corrupted and missing data instances due to instrumentation limitations were excluded from further analysis. Xiao and Fan [10] addressed that missing values can be filled in using the global constant, moving average, or inference-based models. In their paper, missing values were handled using a simple moving average method with a window size of five samples. Rahman et al. [11] proposed an imputation scheme using a recurrent neural network model to provide missing values in time series energy consumption data. The missing-value imputation scheme has been shown to obtain higher accuracies than those obtained using a multilayer perceptron model. Peppanen et al. [12] presented a novel and computationally efficient data processing method, called the optimally weighted average data imputation method, for imputing bad and missing load power measurements to create full power consumption data sets. The imputed data periods have a continuous profile with respect to the adjacent available measurements, which is a highly desirable feature for time-series power flow analyses. Ma et al. [13] proposed a methodology called the hybrid Long Short-Term Memory model with Bi-directional Imputation and Transfer Learning (LSTM-BIT). It integrates the powerful modeling ability of deep learning networks and flexible transferability of transfer learning. A case study on the electric consumption data of a campus lab building was utilized to test the method.

To summarize the above research, researchers introduced, developed, and applied a specific imputation method applied for specific sensor types and missing data scenarios. However, normally in one building system, there are various types of sensors, including a thermometer, humidity sensor, flow meter, energy meter, differential pressure sensors, etc. There are also multiple missing data mechanisms (malfunction, power outage, local interferences, wireless radio transmissions error) and multiple randomness for missing data (missing not at random, missing at random, missing completely at random) for various sensors. The single imputation method may not be able to capture various missing data scenarios from various sensors. Few researchers study the compatibility of a single technique on various sensors with missing data. A few papers are found to work on the selection and use of multiple data imputation methods and they are introduced in the following paragraph.

Inman et al. [14] explored the use of missing data imputation and clustering on building electricity consumption data. The objective was to compare two data imputation methods: Amelia multiple imputation and cubic spline imputation. The results of this study suggest that using multiple imputation to fill in missing data prior to performing clustering analysis results in more informative clusters. Schachinger et al. [15] focused on improving and correcting the monitored data set. They tried to identify the reasons for data losses and then look for recovery options. If lengths of periods of missing data are not exceeding the predefined thresholds, these periods are interpolated. Interpolation thresholds are set depending on the frequency of sensor monitoring and dynamic behavior. Continuous data are interpolated using linear and polynomial interpolation. Habib et al. [16] filled the missing values with regression and linear interpolation for short and long periods. They also addressed that (a) it is necessary to detect the missing gaps between the data acquired as it is also an important factor to indicate the reliability of the data, (b) these missing gaps are always undeniable and lowering the amount of meaningful calculations, (c) it is crucial to make the missing gaps fewer as data with fewer gaps is considered good quality data, and (d) there are different methods available for handling the missing values, e.g., regression, depending on the nature of the data and other parameters, such as computations, precision, robustness, and accuracy. Xia et al. [17] addressed that to get the data in workable order for calculation, analysis, and benchmarking, the missing or invalid (mainly negative) data should be replaced with data during time periods or days that were similar to the invalid points, taking weather condition into account as well. For example, a few missing data would be replaced by the previous or following few proper data, or their average. Several hours’ missing data would be replaced by data of the same time periods on the previous or following day, considering weekdays and weekends. The same goes for missing or invalid data of an even longer period. Garnier et al. [18] used data anterior and/or posterior to the sensor failure in order to rebuild the missing information. Two different approaches were presented: an interpolation technique used for the estimation of missing solar radiation data and an extrapolation technique based on artificial intelligence deals with indoor temperature estimation.

The papers in the last paragraph addressed the selection among only a few (two or three) imputation methods according to the length of missing data gaps, sensor types, and the nature of missing data. Too much domain knowledge is set to select among the sensors, making the method less automated. Ensemble method, which is a pool-based method that can better address the variation problem is rarely applied for building sensors. Additionally, few studies are found to apply the ensemble method to customize the imputation for different sensors with missing data.

One more key issue for data imputation techniques lies in the difficulty of validation. Since the information of missing data are missed already and the truth data is lost forever, it is hard to create some testing or validation scenarios to quantify the performance of data imputation methods.

In this paper, a framework is proposed to deal with the above two problems. First, a validation module is developed based on pattern recognition. This module identifies good data points that have similar characteristics with the missing data. The selected good data can mimic missing data but with truth value, which create testing scenarios to validate the effectiveness of the single data imputation method. Second, with the validation data decided, a pool of data imputation methods is tested under the validation dataset, to find an optimal imputation method for each sensor, which is basically an ensemble method. The selected imputation method is expected to be different from sensor to sensor, which can optimize the accuracy of imputation under different sensor types and different mechanism and randomness of data missing for each sensor.

The paper is organized as follows. The data imputation framework is developed in Section 2. The developed framework is applied to a real-building case study to demonstrate its effectiveness in Section 3. Results and discussion are introduced in Section 4. Conclusions are drawn in Section 5.

## 2. Pattern-Recognition-Based Ensemble Data Imputation Framework

The data imputation framework addresses two problems summarized in Section 1: (1) the lack of validation data to test imputation methods, and (2) the lack of method to automatically customize the imputation method for various sensors with various mechanism and randomness of missing data. Correspondingly, there are two modules to solve the problem, respectively, which are: the validation data generation module (Module 1) and the ensemble imputation module (Module 2). The diagram of the framework and the two modules are shown in Figure 1.

### 2.1. Module 1. Validation Data Generation

As shown in Figure 1, the original timeseries data have good values (green square) and missing data (red square). Module 1 creates validation data (yellow square) using pattern recognition technique. Data with good values (e.g., x_7_ and x_8_ in Figure 1) and the generated validation data (x’_7_ and x’_8_) are the same in terms of values; their only difference is that the validation data are used to evaluate the imputation method. Pattern recognition is applied to identify some good data points that have similar characteristics with the missing data (red square) and marks them as validation data. The characteristics include time indicators (such as hour of day, day of week, weekday weekend indicator, and month of year) and weather variables (such as outdoor air temperature, humidity, and solar radiation). This means that the pattern recognition technique tries to search among good data and find the ones that have similar happening time and weather condition with the missing data, trying to mimic the missing data mechanism. Similarity metrics such as Euclidean distance are used to quantify how missing data and good data are similar with each other. To be more specific, the problem can be formulated and simplified using the equation representing that the method tries to search among all good data ($X_{g}$) to select several good data points ($x_{g}$, a matrix with time and weather variable) with the highest similarity metrics (defined by $f_{sim}$), meaning they are the most similar to the missing data set ($X_{m}$). The selected good data points are supposed to be more likely to miss values. Those data sets that are similar to missing data but have a truth value can be further used as validation datasets to test the effectiveness of data imputation methods in the next steps. The pattern recognition techniques can be unsupervised clustering techniques (such as K-means clustering, categorical mixture models, hierarchical clustering, correlation clustering, and kernel principal component analysis), supervised classification techniques (such as neural network, support vector machine, decision tree and random forest), and any other statistical methods that can classify and cluster data.
(1)$$\underset{x_{g}\;\in \;X_{g}\;}{\mathrm{argmax}}f_{sim}\left(x_{g},X_{m}\right)$$

### 2.2. Module 2. Ensemble Imputation Method

After deciding the validation dataset in Module 1, Module 2 applies the ensemble method to find the data imputation method with the best validation accuracy among a pool of imputation methods for each sensor using the validation dataset. First, it creates a pool of data imputation methods including statistical and model-based methods, for example simple (e.g., nearest available value), linear (e.g., linear interpolation), nonlinear (e.g., quadratic interpolation), piecewise (e.g., piecewise polynomial interpolation), and machine-learning-based interpolation (e.g., support vector machine). Since it is an ensemble method whose performance depends on the method with the best performance, the more diversified and more imputation methods are in the pool, the better performance this ensemble method can achieve. Then, it tests each imputation method using the validation dataset generated in Module 1 and selects the method with the highest validation accuracy from the data imputation method pool for each sensor. The validation accuracy is quantified by some accuracy and error metrics (such as coefficient of variation of root mean squared error and r-squared) between the imputed value and the truth value of the validation data. Finally, all missing data are imputed by the selected data imputation method (blue square shown in Figure 1).

The developed framework can work if there is archiving system to store and manage history data. There is no additional cost on installation. The only cost to apply this framework is the hardware cost for computation. The output of the framework is post-processed and repaired missing data. It can be used for both online and offline missing data imputation, but, in this paper, only an offline process is considered. To demonstrate the application and the effectiveness of the framework, it is applied to the sensors from a real campus building in the next section.

## 3. Real-Building Case Study

### 3.1. Sensors from a Real Campus Building

The sensors from Nesbitt Hall are used to demonstrate the developed framework. Nesbitt Hall (Figure 2) is a campus building located at Drexel University in Philadelphia. It is a seven-story, 7246 m^2^ commercial building that includes classroom, office, and laboratory space. It has one chiller and two steam-to-hot-water heat exchange systems. It has three air handling units (AHUs) located in the basement and on the roof. The chiller system is in the basement with the corresponding cooling tower on the top of the building. More details about heating, ventilation, and air conditioning (HVAC) systems and equipment of Nesbitt Hall can be found in Appendix A.

Among the 539 sensors from Nesbit Hall, 18 sensors with missing data listed in Table 1 are selected to test the framework. There are five considerations of choosing these sensors for the case study: (1) the sensors have adequate missing data; (2) the sensors are typical sensors in BAS; (3) the sensors cover various sensor types, including an energy meter, flow meter, thermometer, humidity sensor, ampere meter, valve position, and differential pressure sensors; (4) the sensors cover various equipment, including a chiller, AHU, and a variable-air-volume (VAV) box, and (5) the sensors have various patterns and characteristics of missing data, including missing data size and randomness. The data are downloaded via the Nesbitt Hall BAS web interface from 1st January to 16th August in 2017, which covers summer, winter, and shoulder seasons. Data are collected every 5 min (i.e., with a 5-min sample interval).

### 3.2. Missing Data Description of Selected Sensors

In Section 3.2, missing data characteristics, including missing data size and missing data randomness, are introduced. In terms of missing data size, Table 2 shows the total number of missing data and maximum holes (or gaps, meaning length of consecutive missing data) for each sensor. Table 2 indicates a varieties of missing data size for the selected sensors.

Most sensor data listed in Table 1 are systematically missed. The best example is the chiller energy meter. Figure 3 shows the Kernel Density Estimation plot reflecting density distribution of missing data happening time (hour of day). It shows that data are more frequently missed during the chiller’s start-up time (around 7–8 am) of the HVAC system, where energy suddenly increases. This indicates that the missing mechanism of chiller energy meter data is somehow systematic and not random. In this study, the systematic missing of each sensor are quantified by the randomness of time indicators for the time when data are missing, including hour of day, day of week, month of year, and weekday indicator (0 for weekday and 1 for weekend). For example, the randomness for the energy meter in terms of hour of day is low because data are missed following a pattern of fault happening time, or data are systematically missing in terms of hour of day in energy meters. We use a statistical test to quantify the randomness of missing data in this study. The null hypothesis is that the data are generated in a random manner, and the equation of randomness is shown in Equation (1), which can be found in the book [19]. In Equation (2), R is the number of runs. The concept of run is defined as a series of consecutive positive (or negative) values of the difference between one data point and the median of all data. n1 is the number of data that is greater than the median of all data, and n2 is the number of data that is less than the median.
(2)$$Z=\;\frac{R-\;\frac{2n_{1}n_{2}}{n_{1}+\;n_{2}}\;-1\;}{\sqrt{\frac{2n_{1}n_{2}\left(2n_{1}n_{2}-n_{1}-\;n_{2}\right)\;}{{\left(n_{1}+\;n_{2}\right)}^{2}\left(n_{1}+\;n_{2}-1\right)}}},$$

The calculated Z-score indicates the randomness of a series of data: under a 10% significance level, the Z-score over 1.65 indicates non-randomness of data. The results of missing data randomness quantification by Z-score by sensor and time indicator are shown in Table 3. By comparing the calculated Z-score, missing data in sensors such as “chws_control/chw_flow” and “peco_meter/elec_usage_chiller” are more systematically missed, while sensors such as “ahu1/htg_stm_vlv_1_3” and “ahu1/stat_press” are less systematically missed. To compare the average of Z-scores among the time indicators, the missing mechanism seems less random and more correlated with hour of day. In other words, the missing data in most sensor are missed under some hourly pattern. It is worth mentioning that this paper will not study the truth of the mechanism of missing data in each sensor but only describe general randomness using Z-score. Moreover, the randomness test is not part of the framework, but it will help readers to better understand the characteristics of missing data for each sensor in the case study.

### 3.3. Create Validation Dataset

A random-forest-based pattern recognition technique is applied to the case study to create the same number of validation data (or “fake” missing data) from good data, by mimicking the real missing data characteristics. Pattern recognition here means we want to recognize the pattern of real missing data to find some good data with similar characteristics with real missing data, so that we can use those good data as validation dataset to test various data imputation methods.

We apply a supervised random-forest-based pattern recognition technique to realize the generation of validation dataset, or “fake” some missing data but with truth value. For each sensor, we label the missing data with 1 and good data with 0, which is the output of a random forest regressor. The inputs for the regressor are weather and time indicators, including outdoor air temperature, outdoor humidity, outdoor air enthalpy, hour of the day, day of the week, weekday indicator (0 for weekday and 1 for weekend), and month of the year. All the information from inputs are from timestamp of sensors and the corresponding weather that is recorded at the timestamp. We train the regressor based on the inputs and output. The model can predict under what time and weather the sensor will possibly miss a value. With that trained model, test the training inputs again and check the value of the output. The larger the testing output (or closer to 1), the more likely the data has the characteristics of missing data in terms of time and weather. Besides the n real missing data from one sensor, selected the other n good data with the highest testing output, which will show similar characteristics with the real missing data. Figure 4 shows an example of how a validation data point is generated based on its similar characteristics with one of the missing data in the chiller energy meter. The random forest algorithm is realized by a Python machine-learning library called scikit-learn. The parameters of random forest algorithm are all set to default.

### 3.4. Pool of Data Imputation Methods

As introduced in Section 2, the developed framework applies the ensemble method for missing data imputation, where multiple imputation methods are used as candidate methods to test the performance on the validation dataset. The method with the best validation accuracy will be the final imputation method for a specific sensor Table 4. lists a pool of the candidate data imputation methods for the real-building case study. The ensemble method is independently conducted for each sensor, meaning that one single imputation method will be selected for each sensor. As can be seen from Table 4, there are five categories of the imputation methods in the pool for the case study: (1) simple interpolation using the next, last, or nearest valid observation to fill the missing data gap (bill, ffill, and nearest); (2) linear interpolation (linear); (3) nonlinear interpolation (quadratic and cubic); (4) piecewise splines (piecewise_polynomial, from_derivatives, pchip, akima); and (5) machine-learning-based interpolation techniques (artificial neural network, or ANN). These data imputation methods are realized by Python libraries including pandas, scipy, and sklearn. It is worth mentioning that the parameters of each method from Python libraries are default and untuned in the case study. The case study is just a showcase or example of the ensemble method with a different single imputation algorithm. The single algorithms can be tuned to get better performance, and chances are that the ensemble method can also achieve better performance accordingly. However, the focus of this paper is not tuning of single algorithm so only default parameters are considered. In the next section, the results and discussion of applying the developed framework on the real-building case study are introduced.

## 4. Results and Discussion

With the developed framework introduced in Section 2 and real-building case study and experiment settings in Section 3, we will implement the developed framework to the case study and discuss the results in Section 4.

The framework generates a set of validation data from each sensor listed in Table 1 according to the real missing data characteristics and pattern recognition techniques introduced in Section 3.3. Then, for each sensor, multiple data imputation methods listed in Table 4 (Section 3.4) are tested under the generated validation dataset. The performance of the data imputation method is quantified by the coefficient of variation of root mean squared error (CVRMSE) of the imputed value calculated by each imputation method and the truth value from the validation dataset. The smaller the error, the better the performance of imputation. The results of imputation performance are summarized in Table 5.

Several conclusions can be drawn by comparing the imputation accuracy among various sensor types and imputation methods from Table 5:
The listed imputation methods have vastly different performance (imputation accuracy). In terms of sensor average, the best imputation method, the developed ensemble method can outperform the worst (bfill) method by 70.0%. The developed ensemble method can increase the accuracy of imputation by 57.1% on average compared with single imputation method. It can also increase the accuracy of imputation by the best single imputation method (ANN in this case) by 18.2%. The results show the effectiveness of improving imputation accuracy and indicate the necessity of the pool-based ensemble imputation methods developed in the framework.The performance of single imputation methods differs from sensor to sensor. Here, single imputation methods refer to all imputation methods that are not the ensemble imputation method. Some single imputation methods can have high accuracy for one sensor but will loses accuracy when used for another. For example, ANN outperforms other single imputation methods on average, but it is worse than pchip when imputing the sensor “vv-1-b-1/air_flow/flow_input”. Moreover, the best single imputation method selected is different from sensor to sensor.The difficulty of sensor prediction is different from sensor to sensor. The fan sensor “ct_control/ct_fan_status/trend_log” has a relatively low testing accuracy with all the imputation methods in the pool, while sensors such as “chws_control/chws_temp” can be easily imputed even with the simplest imputation methods.To further analyze the single imputation method performance in terms of sensor types, the machine-learning-based imputation method is most suitable for the energy meter, and piecewise-spline imputation methods are suitable for some temperature sensors and some pressure sensors. Although the above conclusion may be biased to the specific building and sensors, the result indicates the importance of ensemble method to automatically customize the selection of imputation method based on different sensor types and missing data characteristics.To further analyze the result in terms of the five imputation types, machine-learning-based imputation generally has the best overall performance. Besides, machine-learning methods have even better performance improvement, especially when there are large holes in the missing data, such as the sensor “chws_control/cw_vlv_fdbk”. However, the ensemble method can still always select the best performance among all candidate single imputation methods.

To sum up, the results show that different single imputation methods are suitable for different sensor types and missing data characteristics. The results also show that the proposed framework can optimize the accuracy of imputation under different mechanisms and the randomness of data missing for different sensors. The ensemble imputation method outperforms the best single imputation method by 18.2% in the case study. The result also indicates the importance of the ensemble method to automatically customize the selection of the imputation method based on different sensor types and missing data characteristics.

## 5. Conclusions

This paper developed a framework of data imputation for sensors from building energy systems. The first module of this framework is developed based on pattern recognition, which identifies good data points that have similar characteristics with the missing data. The selected good data can mimic missing data but with truth value, which create testing scenarios to validate the effectiveness of the data imputation method. In the second module of this framework, a pool of data imputation methods is tested under the validation dataset to find an optimal imputation method for each sensor, which is termed as an ensemble method. The selected imputation method is expected to vary from sensor to sensor, which can reflect the specific mechanism and randomness of missing data from each sensor.

The effectiveness of the framework is demonstrated in a real-building case study. The results show the importance of the ensemble method to automatically customize the selection of the imputation method based on different sensor types and missing data characteristics, after finding in the case study that (1) the ensemble method outperforms the best single imputation method by 18.2% and (2) the single imputation method cannot achieve good performance in all types of sensors and missing data characteristics.

The framework can automate the data cleaning process due to very little domain knowledge being required. Users of this framework do not need any knowledge on the characteristics of the sensor and its missing data. Instead, the pool-based ensemble imputation method can automatically evaluate each single imputation method and find the most suitable one for each sensor. The automation process is essentially important with the development of BAS and Internet of Things, where a high automation is required for building analysis, modeling, and control.

In terms of the limitation and future work, first, the framework only considers offline missing data. In the future, it can be extended for an online process. Second, the pool of imputation methods does not exhaust all the existing imputation methods. More sophisticated data imputation methods, such as Amelia, Self-Organization Maps, and K-nearest neighbors, can be included into the pool of imputation methods to further improve the performance of the whole framework. Moreover, the frequency of readings in this paper is 5 min for all sensors; in the future, the impact of sensor reading frequency on the accuracy of the developed framework can be further studied. Finally, the validation data generation module will not find appropriate (or similar enough) validation data when the missing data is a very rare case (in terms of weather and happening time) or no such pattern appears in the dataset. In the future, the impact of this situation on imputation accuracy should be evaluated. Generally speaking, the future work should focus on extending the developed framework to a more generic and plug-n-play tool for BAS.

## Figures and Tables

**Figure 1 sensors-20-05947-f001:**
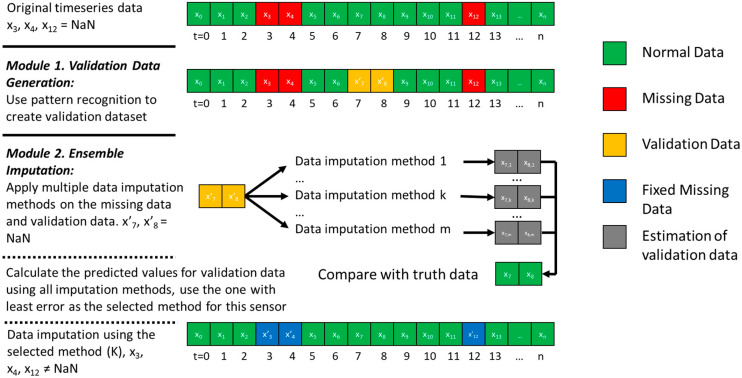
Diagram of the developed data imputation framework with two modules.

**Figure 2 sensors-20-05947-f002:**
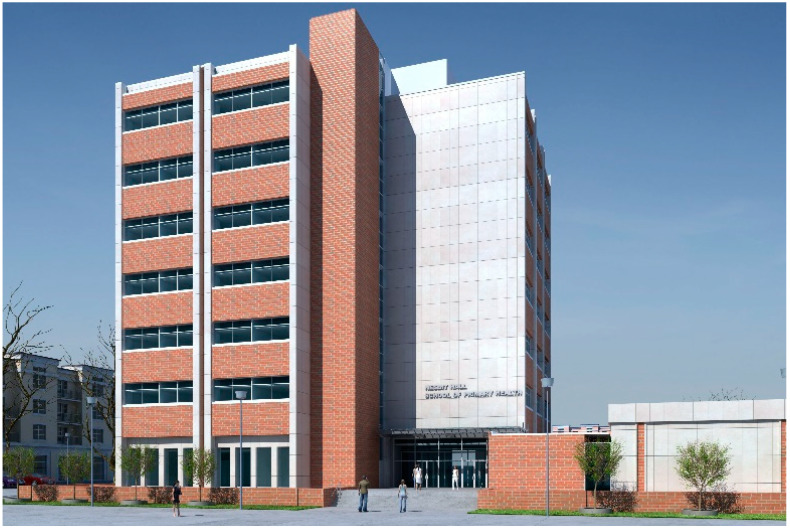
Appearance of Nesbitt Hall.

**Figure 3 sensors-20-05947-f003:**
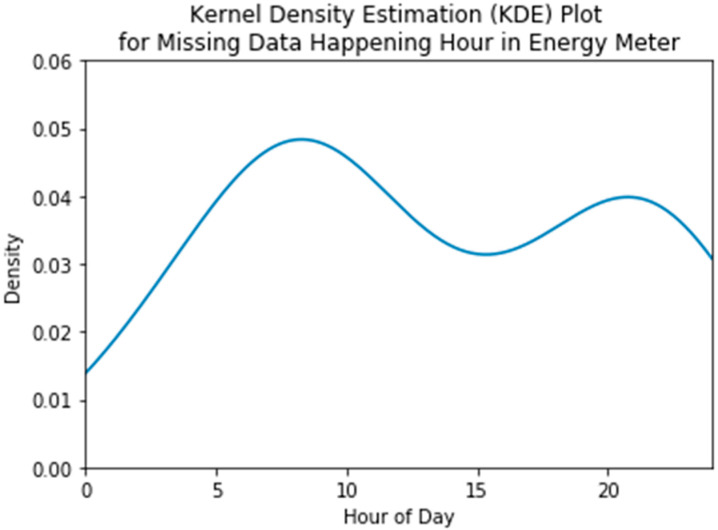
Example of missing data for whole building energy meter.

**Figure 4 sensors-20-05947-f004:**
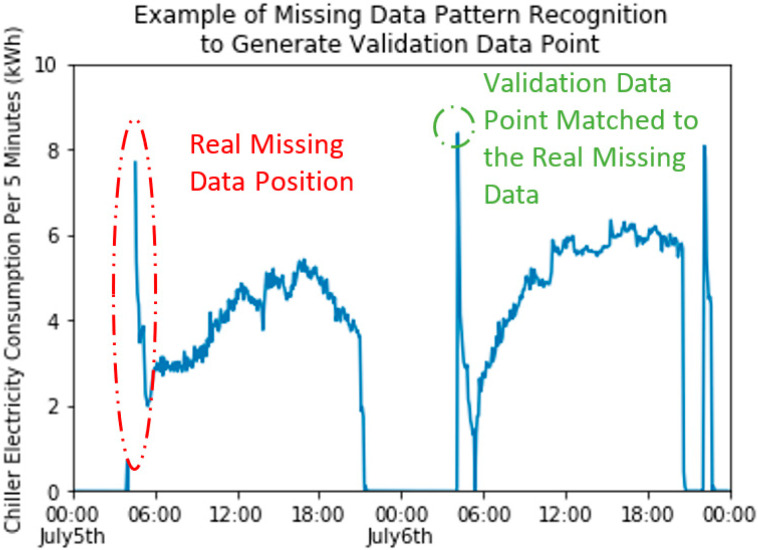
Example of one real missing data point and one generated validation data point in the sensor chiller energy meter.

**Table 1 sensors-20-05947-t001:** Sensors used in the case study of the developed framework.

No.	Equipment	Sensor Type	Original Sensor Name from BAS	Sensor Description
1	Chiller	Energy meter	peco_meter/elec_usage_chiller	Chiller electricity consumption
2		Flow meter	chws_control/chw_flow	Chilled water flow rate
3		Thermometer	chws_control/chws_temp	Chilled water supply temperature
4		Differential pressure sensors	chws_control/delta_press	Chilled water supply and return pressure difference
5	Cooling tower	Differential pressure sensors	chws_control/cw_delta_press	Cooling water supply and return pressure difference
6		Thermometer	chws_control/cwr_temp	Cooling water return temperature
7		Valve position	chws_control/cw_vlv_fdbk	Cooling water valve feedback
8		Fan status	ct_control/ct_fan_status/trend_log	Cooling tower fan status
9	AHU	Thermometer	ahu1/ra_temp/trend_log	AHU 1 return air temperature
10		Valve position	ahu1/econ	AHU 1 economizer valve position
11		Valve position	ahu1/chw_valve	AHU1 cooling coil valve position
12		Humidity sensor	ahu1/ra_humidity	AHU1 return air humidity
13		Ampere meter	ahu1/sfan_amps	AHU 1 supply air fan ampere
14		Valve position	ahu1/htg_stm_vlv_1_3	AHU 1 steam valve position
15		Differential pressure sensors	ahu1/stat_press	AHU 1 supply air static pressure
16		Thermometer	ahu1/pht_temp	AHU 1 preheat coil water temperature
17	VAV	Flow meter	vv-1-b-1/air_flow/flow_input	VAV 1 air flow rate
18		Thermometer	vv-1-b-1/da_temp/trend_log	VAV 1 discharge air temperature

**Table 2 sensors-20-05947-t002:** Missing data description of the selected sensors.

No	Sensor Name	Number of Missing Data	Maximum Hole Size
1	peco_meter/elec_usage_chiller	466	2
2	chws_control/chw_flow	512	2
3	chws_control/chws_temp	439	1
4	chws_control/delta_press	505	2
5	chws_control/cw_delta_press	236	2
6	chws_control/cwr_temp	446	1
7	chws_control/cw_vlv_fdbk	531	3
8	ct_control/ct_fan_status/trend_log	426	1
9	ahu1/ra_temp/trend_log	466	1
10	ahu1/econ	269	2
11	ahu1/chw_valve	315	3
12	ahu1/ra_humidity	446	2
13	ahu1/sfan_amps	334	2
14	ahu1/htg_stm_vlv_1_3	118	2
15	ahu1/stat_press	118	1
16	ahu1/pht_temp	236	2
17	vv-1-b-1/air_flow/flow_input	216	3
18	vv-1-b-1/da_temp/trend_log	571	1

**Table 3 sensors-20-05947-t003:** Randomness of missing data (Z-score) for each sensor in terms of hour of day, month, and weekday indicator.

Sensor Name	Hour	Day	Weekday Indicator	Month	Average
peco_meter/elec_usage_chiller	2.30	1.17	2.06	2.18	**1.93**
chws_control/chw_flow	2.26	0.79	2.01	2.15	**1.80**
chws_control/chws_temp	2.09	0.22	1.86	1.96	1.53
chws_control/delta_press	2.24	0.71	1.99	2.12	1.77
chws_control/cw_delta_press	1.53	0.57	1.26	1.35	1.18
chws_control/cwr_temp	2.11	0.27	1.80	1.97	1.54
chws_control/cw_vlv_fdbk	2.15	0.68	1.88	2.02	1.69
ct_control/ct_fan_status/trend_log	2.06	0.18	1.78	1.92	1.49
ahu1/ra_temp/trend_log	2.15	0.25	1.82	2.02	1.56
ahu1/econ	1.63	0.60	1.35	1.46	1.26
ahu1/chw_valve	1.77	0.82	1.41	1.61	1.40
ahu1/ra_humidity	2.11	0.68	1.83	1.99	1.65
ahu1/sfan_amps	1.82	0.55	1.43	1.67	1.37
ahu1/htg_stm_vlv_1_3	1.08	0.36	0.61	0.86	**0.73**
ahu1/stat_press	1.08	0.01	0.59	0.82	**0.62**
ahu1/pht_temp	1.53	0.56	1.17	1.35	1.15
vv-1-b-1/air_flow/flow_input	1.46	0.70	1.18	1.27	1.15
vv-1-b-1/da_temp/trend_log	2.39	0.39	2.14	2.27	1.79
Average	**1.85**	0.49	1.54	1.70	1.39

**Table 4 sensors-20-05947-t004:** Pool of data imputation methods for the case study.

No.	Category	Abbreviation	Definition and Description	Library or Reference
1	Simple	bfill	use next valid observation to fill gap	pandas.DataFrame.fillna
2	Simple	ffill	use last valid observation to fill gap	pandas.DataFrame.fillna
3	Simple	nearest	Use the nearest available value	scipy.interpolate.interp1d
4	Linear	linear	Linear interpolation	pandas.DataFrame.interpolate
5	Nonlinear	quadratic	Interpolation of second order	scipy.interpolate.interp1d
6	Nonlinear	cubic	Interpolation of third order	scipy.interpolate.interp1d
7	Spline	piecewise_polynomial	Piecewise polynomial curve specified by points and derivatives	scipy.interpolate.PiecewisePolynomial
8	Spline	from_derivatives	Piecewise polynomial in the Bernstein basis	scipy.interpolate.BPoly.from_derivatives
9	Spline	pchip	Piecewise Cubic Hermite Interpolating Polynomial	scipy.interpolate.PchipInterpolator
10	Spline	akima	Akima spline interpolation	scipy.interpolate.Akima1DInterpolator
11	Machine learning	ANN	Use Multi-layer Perceptron to predict the missing value based on weather and time	sklearn.neural_network.MLPRegressor

**Table 5 sensors-20-05947-t005:** Summary of the performance (coefficient of variation of root mean squared error (CVRMSE)) of imputation methods for each sensor.

Sensors\Imputation Methods	bfill	ffill	ANN	Linear	Nearest	Quadratic	Cubic	pp *	Fd *	pchip	akima	Avg *	Ensemble
peco_meter/elec_usage_chiller	0.03	0.04	0.02	0.02	0.03	0.04	0.04	0.02	0.02	0.02	0.02	0.03	0.02
chws_control/chw_flow	0.43	0.33	0.14	0.23	0.27	0.26	0.27	0.23	0.23	0.23	0.24	0.26	0.14
chws_control/chws_temp	0.03	0.03	0.01	0.02	0.03	0.02	0.02	0.02	0.02	0.02	0.02	0.02	0.01
chws_control/delta_press	0.83	0.50	0.30	0.37	0.48	0.26	0.27	0.37	0.37	0.29	0.30	0.39	0.26
chws_control/cw_delta_press	0.07	0.14	0.02	0.05	0.04	0.04	0.05	0.05	0.05	0.04	0.04	0.05	0.02
chws_control/cwr_temp	0.01	0.01	0.01	0.01	0.01	0.02	0.02	0.01	0.01	0.01	0.01	0.01	0.01
chws_control/cw_vlv_fdbk	0.49	0.49	0.16	0.44	0.49	0.66	0.68	0.44	0.44	0.45	0.47	0.47	0.16
ct_control/ct_fan_status/trend_log	1.94	1.82	0.53	1.58	1.87	2.03	2.08	1.58	1.58	1.62	1.65	1.66	0.53
ahu1/ra_temp/trend_log	0.01	0.01	0.01	0.00	0.01	0.00	0.00	0.00	0.00	0.00	0.00	0.01	0.00
ahu1/econ	0.47	0.09	0.16	0.16	0.07	0.16	0.18	0.16	0.16	0.13	0.13	0.17	0.07
ahu1/chw_valve	0.26	0.28	0.11	0.12	0.06	0.19	0.20	0.12	0.12	0.09	0.09	0.15	0.06
ahu1/ra_humidity	0.04	0.04	0.02	0.02	0.03	0.02	0.02	0.02	0.02	0.02	0.02	0.03	0.02
ahu1/sfan_amps	0.13	0.16	0.09	0.08	0.09	0.08	0.08	0.08	0.08	0.08	0.08	0.09	0.08
ahu1/htg_stm_vlv_1_3	0.16	0.10	0.06	0.10	0.08	0.10	0.12	0.10	0.10	0.09	0.10	0.10	0.06
ahu1/stat_press	0.06	0.13	0.09	0.06	0.13	0.16	0.17	0.06	0.06	0.07	0.07	0.10	0.06
ahu1/pht_temp	0.03	0.01	0.02	0.01	0.00	0.01	0.02	0.01	0.01	0.01	0.01	0.01	0.00
vv-1-b-1/air_flow/flow_input	0.44	0.52	0.26	0.23	0.43	0.23	0.25	0.23	0.23	0.21	0.25	0.30	0.21
vv-1-b-1/da_temp/trend_log	0.01	0.01	0.01	0.01	0.01	0.01	0.01	0.01	0.01	0.01	0.01	0.01	0.01
Avg *	0.30	0.26	0.11	0.19	0.23	0.24	0.25	0.19	0.19	0.19	0.19	0.21	0.09

* pp: piecewise_polynomial, fd: from_derivatives, avg: average.

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
