# Peer review of "A Pattern-Recognition-Based Ensemble Data Imputation Framework for Sensors from Building Energy Systems"

_sensors, 2020, doi:10.3390/s20205947_

Round 1

Reviewer 1 Report

I recommend that the author expand the conclusions with a broader description of future research.

Author Response

Response to Reviewer 1

I would like to thank the reviewer and the editor for your time and feedback.  The constructive comments allowed me to edit, reorganize, restructure, and strengthen the paper. I have endeavored to address your comments either with changes to the manuscript. I will further clarify if I have misinterpreted your comment.

Comment 1. I recommend that the author expand the conclusions with a broader description of future research.

Response:  Thank you for pointing out the issue of a lack of a broader description of future research. To respond to your comment, the following contents about future work are added to Paragraph 1 Section 2.2: “Moreover, the frequency of readings in this paper is 5 minutes for all sensors; in the future, the impact of sensor reading frequency on the accuracy of the developed framework can be further studied. Finally, the validation data generation module will not find appropriate (or similar enough) validation data when the missing data is a very rare case (in terms of weather and happening time) or no such pattern appears in the dataset. In the future, the impact of this situation on imputation accuracy should be evaluated. Generally speaking, the future work should focus on extending the developed framework to a more generic and plug-n-play tool for building automation system.”

Reviewer 2 Report

Review of  A Pattern-Recognition-Based Ensemble Data  Imputation Framework for Sensors from Building Energy Systems

This manuscript presents the framework to address the lack of customized and automated imputation  methodology, and the difficulty of the validation of data imputation methods.

This manuscript is well constructed and provides interesting information.

However, minor revision is recommended before it may be accepted, namely:

- In section 2.1 please provide more information about the methodology of pattern recognition applied in this study –especially please provide measurable characteristic of provide additional schema to explain this issue;

… This means that the pattern recognition technique  tries to search among good data and find the ones that have similar happening…. –because this is very generic description;

- Please explain if the frequency of readings influence the accuracy of proposed method;

- Please indicate clearly main cost of its applications; how it will be used in other systems (so, some additional installations should be done or …); At which stage should be applied (already after readout or based on archival value), since BMS or other types of systems have some own archiving systems. – please provide some additional information, since it is not clear for the readers.

- please do not provide conclusions in section 4. Please extend  rather results discussion.

Author Response

Response to Reviewer 2

I would like to thank the reviewer and the editor for their time and feedback.  The constructive comments allowed me to edit, reorganize, restructure, and strengthen the paper. I have endeavored to address your comments either with changes to the manuscript, or, in a few cases, explain how I believe the manuscript already addressed the comment. I will further clarify if I have misinterpreted any comments.

This manuscript presents the framework to address the lack of customized and automated imputation  methodology, and the difficulty of the validation of data imputation methods. This manuscript is well constructed and provides interesting information.However, minor revision is recommended before it may be accepted, namely:

Comment 1. In section 2.1 please provide more information about the methodology of pattern recognition applied in this study –especially please provide measurable characteristic of provide additional schema to explain this issue;

… This means that the pattern recognition technique  tries to search among good data and find the ones that have similar happening…. –because this is very generic description;

Response: This is a good point to clarify the methodology with more quantifiable expressions. I formulated and simplified the problem using equation 1 and add detailed explanation to this problem formulation as shown in Paragraph 1 Section 2.1 : Similarity metrics such as Euclidean distance are used to quantify how missing data and good data are similar with each other. To be more specific, the problem can be formulated and simplified using the following equation,

,

(1)

, representing that the method tries to search among all good data () to select several good data points (, a matrix with time and weather variable) with the highest similarity metrics (defined by ), meaning they are the most similar to the missing data set ().

Comment 2. Please explain if the frequency of readings influence the accuracy of proposed method;

Response: This is a good point to clarify the frequency of readings. The topic of missing data is discussed in the context of typical building automation system, where data resolution is from seconds to minutes. The developed framework is feasible within this range. However in this paper, I didn’t study how the frequency will impact on the accuracy of proposed method, mainly because of the limitation of the data I have for the case study: all data collected from Nesbitt Hall are under the time interval of 5 minutes. I think this topic can be considered as the future work. As a result, the following contents are added to Paragraph 1 Section 2.2: “Moreover, the frequency of readings in this paper is 5 minutes for all sensors; in the future, the impact of sensor reading frequency on the accuracy of the developed framework can be further studied.”

Comment 3. Please indicate clearly main cost of its applications; how it will be used in other systems (so, some additional installations should be done or …); At which stage should be applied (already after readout or based on archival value), since BMS or other types of systems have some own archiving systems. – please provide some additional information, since it is not clear for the readers.

Response: It is great point to clarify when and where does the framework work. There is no extra cost if the BAS has a database to store history data. There is no extra installation. The framework works at the stage of building operation based on archival value. The only cost of this application is computation cost to apply the ensemble algorithm that combining multiple single imputation algorithm. To respond to this comment, the following sentences are added in Paragraph 2 Section 2.2: “The developed framework can work if there is archiving system to store and manage history data. There is no additional cost on installation. The only cost to apply this framework is the hardware cost for computation. The output of the framework is post-processed and repaired missing data. It can be used for both online and offline missing data imputation, but in this paper, only offline process is considered.”

Comment 4. please do not provide conclusions in section 4. Please extend  rather results discussion.

Response: I double-checked the Section 4 and made sure that the bullet points listed in Section 4 are findings, results, or discussions, instead of conclusions. All conclusions are drawn in Section 5.

Reviewer 3 Report

The paper is well prepared with an interesting topic on missing data in building energy. There are several comments: 1. The Validation Dataset Generation method is interesting, which is a random-forest-based pattern recognition technique. However, what if the missing data is a very rare case and no such pattern appears in the dataset? How to handle such cases? 2. In The pool of data imputation methods for the case study, it seems each method is directly used from Python library. However, the parameters of each method should be tuned to pursue a better performance. Thus, the presented results are too simple only using default parameter setting. When parameters are optimized, the result could vary and more analyses are required. 3. The study did not compare the computation of single and ensemble method, which could be a critical issue in applications. 4. The literature review is not enough. More related work should be reviewed.

Author Response

Response to Reviewer 3

I would like to thank the reviewer and the editor for their time and feedback.  The constructive comments allowed me to edit, reorganize, restructure, and strengthen the paper. I have endeavored to address your comments either with changes to the manuscript, or, in a few cases, explain how I believe the manuscript already addressed the comment. I will further clarify if I have misinterpreted any comments.

The paper is well prepared with an interesting topic on missing data in building energy. There are several comments:

Comment 1. The Validation Dataset Generation method is interesting, which is a random-forest-based pattern recognition technique. However, what if the missing data is a very rare case and no such pattern appears in the dataset? How to handle such cases?

Response: It is a very good point in terms of the boundaries of pattern recognition techniques applied in the framework. In terms of the pattern recognition algorithm, it will still find the data that closest to the rare unseen case. However, if in the case that the reviewer mentioned happens, this selected data cannot truly represent the rare case. To be honest, the framework cannot handle this situation where a missing data happens at a very rare weather and time. It is one of the limitations of this method. As a result, I added the following sentence in Paragraph 4 Section 5: “Finally, the validation data generation module will not find appropriate (or similar enough) validation data when the missing data is a very rare case (in terms of weather and happening time), and no such pattern appears in the dataset. In the future, the impact of this situation on imputation accuracy should be evaluated.”

Comment 2. In The pool of data imputation methods for the case study, it seems each method is directly used from Python library. However, the parameters of each method should be tuned to pursue a better performance. Thus, the presented results are too simple only using default parameter setting. When parameters are optimized, the result could vary and more analyses are required.

Response: This is also a great point about how to set up the single imputation method. The case study is just a showcase or example of ensemble method with different algorithm. The single algorithms can be tuned to get better performance, and chances are that the ensemble method can also achieve better performance accordingly. The focus of this paper is not tuning of each algorithm. To pursue higher accuracy, for the same algorithm, people can even use differently tuned parameters as two individual algorithms for in the ensemble method. To respond to this comment, the following sentences are added in Paragraph 1 Section 3.4: “It is worth mentioning that the parameters of each method from Python libraries are default and untuned in the case study. The case study is just a showcase or example of ensemble method with different single imputation algorithm. The single algorithms can be tuned to get better performance, and chances are that the ensemble method can also achieve better performance accordingly. However, the focus of this paper is not tuning of single algorithm so only default parameters are considered.”

Comment 3. The study did not compare the computation of single and ensemble method, which could be a critical issue in applications.

Response: The paper shows the comparison between ensemble method and single method in the first bullet point of Section 4:

  • The listed imputation methods have vastly different performance (imputation accuracy). In terms of sensor average, the best imputation method, the developed ensemble method, can outperform the worst (bfill) method by 70.0%. The developed ensemble method can increase the accuracy of imputation by 57.1% on average compared with single imputation method. It can also increase the accuracy of imputation by the best single imputation method (ANN in this case) by 18.2%. The results show the effectiveness on improving imputation accuracy and indicate the necessity of the pool-based ensemble imputation methods developed in the framework.

Comment 4. The literature review is not enough. More related work should be reviewed. 

Response: More literatures are added. First, some general reviews for data imputation work are summarized in Paragraph 3 Section 1: “In statistics, imputation is the process of replacing missing data with substituted values [5]. The theories of imputation are well studied. Early in 2001, Pigott [6] wrote a review of methods for missing data imputation and he summarized that ad hoc methods, such as complete case analysis, available case analysis (pairwise deletion), or single-value imputation, are widely studied. Ad hoc methods can be easily implemented, but they require assumptions about the data that rarely hold in practice. However, model-based methods, such as maximum likelihood using the expectation-maximization algorithm and multiple imputation, hold more promise for dealing with difficulties caused by missing data. While model-based methods require specialized computer programs and assumptions about the nature of the missing data, these methods are appropriate for a wider range of situations than the more commonly used ad hoc methods. Harel and Zhou [7] reviewed some key theoretical ideas forming the basis of multiple imputation and its implementation, provided a limited software availability list detailing the main purpose of each package, and illustrate by example the practical implementations of multiple imputation, dealing with categorical missing data. Ibrahim et al. [8] reviewed four common approaches for inference in generalized linear models with missing covariate data: maximum likelihood, multiple imputation, fully Bayesian, and weighted estimating equations. They studied how these four methodologies are related, the properties of each approach, the advantages and disadvantages of each methodology, and computational implementation. They examined data that are missing at random and nonignorable missing. They used a real dataset and a detailed simulation study to compare the four methods.”

Second, I add one new paper that applies single data imputation method published in 2020 in Paragraph 4 Section 1: “Ma et al. [13] proposed a methodology called the hybrid Long Short Term Memory model with Bi-directional Imputation and Transfer Learning (LSTM-BIT). It integrates the powerful modeling ability of deep learning networks and flexible transferability of transfer learning. A case study on the electric consumption data of a campus lab building was utilized to test the method.”

Reviewer 4 Report

Strong and well described research.  The findings provide a valuable contribution that can be widely applied.  Several reccomendations include:

  • While figure 1 is very clear and helpful…. with regard to Module 1,  it would be helpful to have more information about how x7’ and x8’ differs from x7 and x8.  It is clear how x7’ and x8’ are selected (similar time and weather). However, it is not clear (as the notation suggests) how they differ.  This step requires more explanation.
  • Figure 3 does not add significantly. Recommend deletion.
  • While many of the sensors are for indoor equipment and characteristics, which may minimize the following comment… this reviewer wonders if time of year, in addition to time of day (and weather), should be taken into account in the generation of validation data, due to the possibility that sun angles have an impact on results.

In general, the research is well justified and presented and is worthy of publication.   

Author Response

Response to Reviewer 4

I would like to thank the reviewer and the editor for their time and feedback.  The constructive comments allowed me to edit, reorganize, restructure, and strengthen the paper. I have endeavored to address your comments either with changes to the manuscript, or, in a few cases, explain how I believe the manuscript already addressed the comment. I will further clarify if I have misinterpreted any comments.

Strong and well described research.  The findings provide a valuable contribution that can be widely applied.  Several reccomendations include:

Comment 1. While figure 1 is very clear and helpful…. with regard to Module 1,  it would be helpful to have more information about how x7’ and x8’ differs from x7 and x8.  It is clear how x7’ and x8’ are selected (similar time and weather). However, it is not clear (as the notation suggests) how they differ.  This step requires more explanation.

Response: It is a great point about the clarification issue. The following changes are made in Paragraph 1 Section 2.1: “Data with good values (e.g., x7 and x8 in Figure 1) and the generated validation data (x’7 and x’8) are the same in terms of values; their only difference is that the validation data are used to evaluate imputation method. Pattern recognition is applied to identify some good data points that have similar characteristics with the missing data (red square) and marks them as validation data.”

Comment 2. Figure 3 does not add significantly. Recommend deletion.

Response: Figure 3 shows that data are more frequently missed during the chiller’s start-up time (around 7-8 am) of HVAC system, where energy suddenly increases. It is important to show that chiller energy meter data is somehow systematic and not random, and it can also help readers to better understand the following randomness index.

It confuses the reviewer maybe because I originally put Figure 3 below the Paragraph 3 Section 3.2, which is far away from the texts that explain the figure. As a result, I moved the figure 3 below the Paragraph 2 Section 3.2.

Comment 3. While many of the sensors are for indoor equipment and characteristics, which may minimize the following comment… this reviewer wonders if time of year, in addition to time of day (and weather), should be taken into account in the generation of validation data, due to the possibility that sun angles have an impact on results.

In general, the research is well justified and presented and is worthy of publication.   

Response: Thank you for bringing this up. I also thought of sun angles changes with the time of the year. Instead of using time of year, I use “month” the reflect this change. As can be seen from Table 3, hour of day, day of week, weekday indicator, and month of the year are used to define the time.

Round 2

Reviewer 3 Report

I am good with the response. No further comments.